# Impact of COVID-19 Pandemic Restrictions on Respiratory Virus Patterns: Insights from RSV Surveillance in Gwangju, South Korea

**DOI:** 10.3390/v16060850

**Published:** 2024-05-26

**Authors:** Sun-Ju Cho, Sun-Hee Kim, Jeongeun Mun, Ji-eun Yun, Sujung Park, Jungwook Park, Yeong-Un Lee, Ji-su Park, Haebi Yun, Cheong-mi Lee, Jong-Pil Kim, Jung-Mi Seo

**Affiliations:** Division of Emerging Infectious Disease, Department of Infectious Disease Research, Health and Environment Research Institute of Gwangju, Gwangju 61954, Republic of Korea; sunny1989@korea.kr (S.-H.K.); mjo3214@korea.kr (J.M.); jeyun11@korea.kr (J.-e.Y.); sujung9421@korea.kr (S.P.); jwpvet@korea.kr (J.P.); gloryw3@korea.kr (Y.-U.L.); nyoil658@korea.kr (J.-s.P.); haebb2@korea.kr (H.Y.); thefirstlee@korea.kr (C.-m.L.); kjp8498@korea.kr (J.-P.K.); sm6698@korea.kr (J.-M.S.)

**Keywords:** respiratory syncytial virus, non-pharmaceutical interventions, whole-genome sequencing

## Abstract

The social restriction measures implemented due to the COVID-19 pandemic have impacted the pattern of occurrences of respiratory viruses. According to surveillance results in the Gwangju region of South Korea, respiratory syncytial virus (RSV) did not occur during the 2020/2021 season. However, there was a delayed resurgence in the 2021/2022 season, peaking until January 2022. To analyze this, a total of 474 RSV positive samples were investigated before and after the COVID-19 pandemic. Among them, 73 samples were selected for whole-genome sequencing. The incidence rate of RSV in the 2021/2022 season after COVID-19 was found to be approximately three-fold higher compared to before the pandemic, with a significant increase observed in the age group from under 2 years old to under 5 years old. Phylogenetic analysis revealed that, for RSV-A, whereas four lineages were observed before COVID-19, only the A.D.3.1 lineage was observed during the 2021/2022 season post-pandemic. Additionally, during the 2022/2023 season, the A.D.1, A.D.3, and A.D.3.1 lineages co-circulated. For RSV-B, while the B.D.4.1.1 lineage existed before COVID-19, both the B.D.4.1.1 and B.D.E.1 lineages circulated after the pandemic. Although atypical RSV occurrences were not due to new lineages, there was an increase in the frequency of mutations in the F protein of RSV after COVID-19. These findings highlight the need to continue monitoring changes in RSV occurrence patterns in the aftermath of the COVID-19 pandemic to develop and manage strategies in response.

## 1. Introduction

Respiratory syncytial virus (RSV) is the most common cause of lower respiratory tract infections in infants worldwide, with the majority being infected before the age of 2, often experiencing repeat infections [1,2]. While it presents symptoms similar to those of the common cold, in severe cases, it can lead to serious conditions such as bronchiolitis and pneumonia, making it a significant pathogen even in elderly and immunocompromised patients [3]. According to a global surveillance study in 2019, 33 million people are infected with RSV annually, with 10% of them requiring hospitalization, and approximately 3% of hospitalized cases resulting in death [4]. Although the burden of illness related to RSV is significant, there is currently no treatment available to effectively control RSV. Treatment for RSV mostly involves supportive care, with severe cases requiring oxygen therapy and mechanical ventilation [5,6,7]. Thus, at present, the most effective way to control RSV is through prevention. In 2023, vaccines for RSV were approved in the United States, United Kingdom, and Japan [8,9,10,11]. 

RSV is a negative-sense RNA belonging to the family Pneumoviridae, with a single-stranded membrane and a 15 kb genome encoding 11 proteins. Among these, the glycoproteins G and F, present on the virus membrane surface, serve as the major antigenic sites responsible for virus attachment to cells and subsequent fusion with cell membranes, initiating infection [12]. The portion of the G protein excluding its central domain exhibits diverse variations and is targeted for RSV genotyping [13]. However, the F protein is relatively stable with fewer mutations, making it a candidate target for subunit vaccine development [5,14].

RSV can be divided into two subgroups, A and B, based on antigenicity. Each subgroup historically harbors numerous detailed genotypes based on the sequence of the G gene. Research on the molecular genetics of RSV has predominantly focused on the G gene, particularly its second hypervariable region, which exhibits the highest genetic and antigenic variability [15,16,17,18]. In particular, since the confirmation of genotypes containing duplications in the G gene of RSV, namely the 72-nt duplication in the RSV A subgroup and the 60-nt duplication in the RSV B subgroup, these genotypes have been the most prevalent since 2016 [19,20,21,22,23]. Detecting such changes may be challenging with partial G gene analysis alone, potentially leading to overlooking alterations in other regions of RSV. However, whole-genome analysis allows for the comprehensive surveillance of RSV by identifying the entire viral genome, including the F gene targeted by recently approved RSV vaccines and monoclonal antibodies [24,25,26].

RSV typically exhibits a pattern of seasonal outbreaks in temperate climates of the Northern Hemisphere, usually starting in the fall, peaking in winter, and declining in spring [27,28]. However, non-pharmaceutical interventions implemented to prevent the spread of the COVID-19 pandemic, such as border closures, social distancing, and mandatory mask-wearing, have affected the occurrence of seasonal viruses, including RSV [29,30,31,32,33]. In the case of South Korea, during the 2020/2021 season, when non-pharmaceutical interventions were in place, RSV did not occur. After the easing of social distancing measures, RSV occurred with a delayed onset in December compared to in the pre-COVID-19 pandemic seasons.

Understanding the relationship between the delayed outbreak of RSV and the virus’ characteristics is necessary to predict and develop preventive measures against future RSV epidemics. In this study, we conducted a phylogenetic analysis of the whole genome of RSV to analyze the molecular epidemiological characteristics of RSV pre- and post-COVID-19 pandemic.

## 2. Materials and Methods

### 2.1. Surveillance and Sample Collection

The Korea Influenza and Respiratory Virus Surveillance System (KINRESS) is a program operated by the Korea Disease Control and Prevention Agency (KDCA). Under this program, major hospitals nationwide are selected for surveillance sampling. Samples are collected weekly from patients with acute respiratory illnesses and sent to designated research institutions for analysis. We, as a participating research institution in KINRESS, monitor acute respiratory infections (ARIs) in the Gwangju area, which has a population of approximately 1.5 million. From September 2018 to August 2023, a total of 8917 throat or nasopharyngeal swabs from outpatients with ARIs were collected from three collaborating hospitals in the Gwangju area for the diagnosis of acute respiratory infections. Screening was conducted for eight types of acute respiratory viruses, including RSV. Descriptive data were presented as frequency and percentage, and group comparisons were performed using the chi-square test. Results with a *p*-value less than 0.01 were considered significant.

### 2.2. RNA Extraction and Real-Time PCR

Following the manufacturer’s instructions, nucleic acids were extracted from the samples using a QIAamp RNA kit (Qiagen, Hilden, Germany). We used 140 µL of samples and 60 µL of final nucleic acid elutions. RSV was identified by using a One-step RSV A&B/HMPV Real-time PCR Kit (Kogenebiotech, Seoul, Republic of Korea South Korea) in accordance with the manufacturer’s instructions. The amplification conditions were as follows: 50 °C for 30 min, followed by 95 °C for 10 min and 40 cycles of 95 °C for 15 s and 60 °C for 1 min. 

### 2.3. Library Preparation and Sequencing

Among the 474 RSV-positive samples, a total of 73 samples for whole-genome sequencing were randomly selected based on Ct values of ≤25: 15 RSV strains from 2018–2020, and 58 from 2021–2023. The viral RNA was extracted using a QIAamp Viral RNA mini Kit (Qiagen, Hilden, Germany) according to the manufacturer’s manual. Libraries were prepared using the Illumina RNA Enrichment Prep with Enrichment(L) Tagmentation and Viral Surveillance Panel (Illumina, San Diego, CA, USA). The libraries were pooled, cleaned, and quantified using Qubit™ 1X dsDNA HS (High Sensitivity) Assay kits on a Qubit 3 Fluorometer (Thermo Fisher Scientific, Waltham, MA, USA). The final library pool molarity and fragment length distribution were determined using the 4200 TapeStation System with high-sensitivity D5000 tape (Agilent, Santa Clara, CA, USA). Libraries were pooled, denatured, and diluted to 9 pM. Sequencing was performed on a Miseq instrument (Illumina, San Diego, CA, USA) using an Illumina Miseq reagent kit v2 (300 cycles) with dual-indexed paired-end 2 × 151 bp reads.

### 2.4. Sequence Data and Analysis

Raw paired sequences generated from MiSeq were imported into the CLC Genomics Workbench version 21.0.3 (CLC bio, QIAGEN, Aarhus, Denmark) for analysis. Using the CLC program, the raw paired sequences were mapped onto the human genome to extract RSV-specific reads (reads that did not map to the human genome). Reference mapping was conducted, and a de novo assembly workflow was performed to identify the optimal reference for each extracted RSV-specific raw sequence. The generated contigs were then used to select the optimal reference through NCBI’s BLAST (National Center for Biotechnology Information Basic Local Alignment Search Tool). Subsequently, the sequencing reads were aligned and mapped, and consensus sequences were generated through the reference mapping workflow. Consensus sequences with ≥95% coverage at 10× depth of the genome were utilized for analysis. For the analysis of variation in RSV-A, the reference sequence A/England/397/2017 was used, which is available under accession number EPI_ISL_412866 in GISAID. Similarly, for the analysis of variation in RSV-B, the reference sequence B/Australia/VIC-RCH056/2019 was used, available under accession number EPI_ISL_1653999 in GISAID. 

### 2.5. Phylogenetic Analysis

Multiple sequence alignment was performed using the MUSCLE algorithm in Molecular Evolutionary Genetics Analysis 11(MEGA 11 ver. 11.0.13) software. Phylogenetic trees were constructed using the Maximum Likelihood (ML) method with the General Time Reversible model in MEGA 11 software(MEGA 11 ver. 11.0.13). The reliability of the branching order was assessed by performing 1000 bootstrap replicates.

## 3. Results

### 3.1. Epidemiology of RSV

The monthly incidence pattern of RSV before the COVID-19 pandemic, during the 2018/2019 and 2019/2020 seasons, showed a typical temperate region pattern, with cases starting in September, peaking in November during the 2018/2019 season and in December during the 2019/2020 season, and then decreasing in the following spring. During the 2020/2021 season, non-pharmaceutical interventions were implemented to prevent the spread of the COVID-19 pandemic, and our surveillance results showed that RSV was not detected. In the 2021/2022 season, as social distancing measures implemented to prevent the COVID-19 pandemic were relaxed, RSV began to emerge in December 2021, surged dramatically in January to February of the following year, and decreased in March. Unlike before the COVID-19 pandemic, the incidence of RSV during the 2021/2022 season was delayed, the duration of the outbreak was shorter than before the pandemic, but the incidence rate was significantly (about three times) higher. The 2022/2023 season started in August 2022, peaked in September, seemed to decrease in December, but began increasing again from January 2023, peaking in March and declining after May. Unlike previous seasons, the 2022/2023 season exhibited a biphasic pattern. The monthly occurrence pattern as described above is depicted in Figure 1A. Comparing the occurrence rates of RSV subgroup A and B during the study period, the epidemic in the 2018/2019 season was dominated by RSV A. In the 2019/2020 season, although the initial occurrence rate of RSV B was high, it increased during the mid-season, and overall, the occurrence rate of RSV B was slightly higher at 57.4% compared to RSV A at 42.6%. However, the difference was not significant. Following the COVID-19 pandemic, the 2021/2022 and 2022/2023 seasons commonly showed a pattern where RSV A dominated in the early season before transitioning to an RSV B epidemic. As shown in Figure 1B, during the 2021/2022 season, the occurrence rate of RSV A was slightly higher at 59.6% compared to RSV B at 40.4%. In the 2022/2023 season, the occurrence rates of RSV A and RSV B were 49.1% and 50.9%, respectively, showing no difference in occurrence rates between the subgroups.

### 3.2. Demographic Distribution of RSV

In the seasons before the COVID-19 pandemic (2018/2019 and 2019/2020), the incidence rate was found to be significantly higher in the age group from 0 to under 2 years old, similar to reports provided in previous studies. However, in the 2021/2022 season, not only the age group of 0 to under 2 years old, but also children under 5 years old, had significantly higher incidence rates of RSV. In the 2022/2023 season, similar to before the COVID-19 pandemic, the incidence rate of RSV was significantly higher in the age group from 0 to under 2 years old. It was observed that the age groups affected by RSV increased temporarily only in the season of 2021/2022, immediately after the relaxation of social distancing measures (Table 1).

### 3.3. Phylogenetic Analysis of RSV Whole Genome Sequences

For the phylogenetic analysis of RSV whole-genome sequences, this study referred to the research by Goya et al., which proposed a method for classifying RSV based on its whole-genome sequence. The reference was selected from GitHub https://github.com/orgs/rsv-lineages/repositories (accessed on 15 February 2024). A total of 27 references were selected for RSV A, and 28 references for RSV B, for phylogenetic analysis. Based on the method proposed by Goya et al. for classifying RSV using its whole-genome sequence, an analysis of a total of 42 RSV A whole-genome sequences revealed that all seven samples from the 2018/2019 season were classified under the A.D. single lineage. However, in the 2019/2020 season, five samples of RSV A were classified into various lineages including A.D.1, A.D.2.2, A.D.3, and A.D.3.1. After the COVID-19 pandemic, in the 2021/2022 season, all 12 samples of RSV A were classified under the A.D.3.1 single lineage. In the 2022/2023 season, the 18 samples of RSV A were classified into four different lineages: A.D.3., A.D.3.1, A.D.1, and A.D.1.1(Figure 2).

A total of 31 RSV B whole-genome sequences were subjected to phylogenetic analysis. In the 2018/2019 season, three samples of RSV B were classified as B.D.4.1.1. However, samples from the 2019/2020 season were not analyzed due to a lack of suitable samples for whole-genome analysis conditions. After the COVID-19 pandemic, in the 2021/2022 and 2022/2023 seasons, 28 samples of RSV B were classified as B.D.4.1.1 and B.D.E.1(Figure 3).

### 3.4. Deduced Amino Acid Sequence of the RSV F Protein

The F protein of RSV serves as the target for recently approved RSV vaccines and monoclonal antibodies. Therefore, in this study, analysis based on the CLC genomics workbench (version 21.0.3CLC bio, QIAGEN, Aarhus, Denmark) was conducted using RSV A/England/397/2017 and RSV B/Australia/VIC-RCH056/2019 strains as references to analyze the presence of mutations in the F protein. The frequency of mutations in the antigenic sites of the RSV F protein before and after the COVID-19 pandemic is shown in Table 2. Among the total of six antigenic sites of the F protein, we identified two types of amino acid mutations in antigenic sites I and II of RSV A. Out of a total of 42 RSV A full genomes, the S276N amino acid substitution was detected in 3 samples before the COVID-19 pandemic period, and 19 were observed post COVID-19 pandemic. Specifically, during the 2021/2022 season following the COVID-19 pandemic, the S276N amino acid mutation was observed in all 12 samples of the RSV A subgroup. Although the sample sizes before and after COVID-19 were not equal, based on the analyzed samples, the frequency of the S276N mutation in the F protein of RSV A increased from 25% (3/12) to 63.3% (19/30) after the COVID-19 pandemic. The mutation K42R was observed in four cases among thirty samples (13.3%) after COVID-19. RSV B exhibited seven types of amino acid mutations across its six antigenic sites. Among 33 full genomes of RSV B, no amino acid mutations were observed in samples from before the pre-COVID-19 pandemic period. However, during the post-COVID-19 pandemic period, mutations were observed in seven types, forming combinations of three to four mutations. The most frequently observed combination of mutations was R42K, S211N, S190N, and S389P, observed in 12 samples. The next most frequently observed mutation combination was S211N, S190N, and S389P, observed in six samples. Mutations with combinations of E463D, S276N, and S190N were observed in three samples. A mutation with a combination of S389T, S211N, and S190N was observed in one sample. Although the number of pre-COVID-19 samples was limited to three, thus insufficient to explain the mutations in the F protein before and after COVID-19, mutations in the F protein were observed in 22 out of 28 RSV B samples after COVID-19. Notably, mutations in the RSV F protein antigenic site were frequently observed after the COVID-19 pandemic. 

## 4. Discussion

Recent studies have suggested that non-pharmaceutical interventions implemented to control the COVID-19 pandemic had an unintended effect on respiratory virus epidemics [29,30,31,32,33]. In this context, in Gwangju, South Korea, according to KINRESS results, only a small number of respiratory virus occurrences were recorded in 2020. However, in the fall of 2021, parainfluenza 3 (PIV3) emerged, followed by human metapneumovirus (HMPV) in the fall of 2022, both showing atypical epidemics [34,35]. Although RSV did not occur during the 2020/2021 season, in the 2021/2022 season, unlike before the COVID-19 pandemic, it began to emerge in winter instead of fall, showing a delayed pattern intensifying from December to January of the following year. In the 2022/2023 season, it occurred in autumn, similar to pre-COVID-19 pandemic times, seeming to decrease by December of the following year. However, it resurged in March, extending the duration of the epidemic. Therefore, in this study, whole-genome analysis was conducted to investigate if there were any changes in the epidemic patterns of RSV before and after the COVID-19 pandemic, and if they were related to specific genotypes.

Molecular genetic analysis of RSV has mostly focused on the G protein. Variations in the second HVR of the G protein have been consistently observed, making it a major target for RSV phylogenetic analysis [36,37,38]. However, with the recent approval of the RSV vaccine and the targeting of monoclonal antibody therapies, variable F protein has gained relatively less attention [5,14]. Therefore, there is a need to pay attention to variations in the F protein to evaluate vaccine efficacy in the future. In this respect, whole-genome analysis allows for the comprehensive examination not only of the G protein, but also of variations in the F protein.

In this study, we classified the phylogeny of a total of 73 RSV whole-genome sequences based on research findings that referenced the RSV phylogeny classification proposed by Goya et al. [23]. Before the COVID-19 pandemic, various lineages of the RSV A subgroup, ranging from A.D lineages to sub lineages, such as A.D.1, A.D.3, A.D.3.1, and A.D.2.2, were prevalent. However, following the COVID-19 pandemic and the relaxation of social distancing measures in the 2021/2022 season, only the A.D.3.1 lineage prevailed. Subsequently, in the 2022/2023 season, the A.D.1, A.D.3, and A.D.3.1 lineages re-emerged. This reduction in the diversity of the RSV A subgroup, observed for the first time after the COVID-19 pandemic, has also been documented in studies by Eden et al., Yoshioka et al., and Goya et al. [32,39,40]. This suggests that non-pharmaceutical interventions (NPIs) implemented to mitigate the COVID-19 pandemic, such as mandatory mask-wearing, social distancing, and border closures, may have influenced the transmission of the virus, potentially leading to a decrease in diversity.

In the 2022/2023 season, besides the A.D.3.1 lineage, the A.D.3, A.D.1, and A.D.1.1 lineages were observed again. However, the A.D.3.1 lineage remained within the same clade from the 2019/2020 season to the 2022/2023 season, suggesting that it had been circulating in the local community prior to COVID-19. In contrast, samples belonging to the A.D.1 lineage in the 2022/2023 season formed a different clade from the previous A.D.1 lineage and showed association with the 2017 Netherlands strain. Similarly, samples belonging to the A.D.3 lineage formed a single clade and showed association with the 2018 UK strain. Consequently, the A.D.1 and A.D.3 lineages in the 2022/2023 season were classified into different clades from the strains that prevailed before COVID-19. Therefore, it is possible that the A.D.1 and A.D.3 lineages in the 2022/2023 season were newly introduced into the local community after the COVID-19 pandemic.

RSV B predominantly circulated in the form of the B.D.4.1.1 lineage in the pre-COVID-19 2018/2019 season. However, in the post-COVID-19 2021/2022 and 2022/2023 seasons, both the B.D.4.1.1 and B.D.E.1 lineage were prevalent. The B.D.4.1.1 lineage appeared to form a single clade both before and after the COVID-19 pandemic, suggesting that the B.D.4.1.1 lineage, which was prevalent before the COVID-19 pandemic, continued to persist even after the pandemic. Additionally, the B.D.E.1 lineage, which emerged after the COVID-19 pandemic, is considered a subtype of the B.D.4.1.1 lineage, likely derived from the B.D.4.1.1 lineage present in the local community before COVID-19.

Analyzing the epidemic patterns of RSV before and after the COVID-19 pandemic, at the point when social distancing measures were relaxed post-COVID-19, the atypical occurrence of RSV was determined to be either a pre-existing strain within the local community or derived from strains already existing globally. These findings align with previous studies by Yoshioka et al., LaRinda et al., and Goya et al. [39,40,41]. 

Comparing amino acid variations in the RSV F protein before and after COVID-19, it was found that among the RSV A subgroup, amino acid variations occurred in two out of the six antigenic sites. The most frequent variation observed was S276N in antigen site II, with 22 out of 42 occurrences. Particularly noteworthy is that in the 2021/2022 season, the first season of RSV occurrence after the COVID-19 pandemic, all 12 samples of the RSV A subgroup exhibited the S276N amino acid variation. For the RSV B subgroup, variations of seven types were observed across a total of six antigenic sites, forming three to four combinations. The most frequently observed mutation combination was R42K, S211N, S190N, and S389P, which was observed in 12 samples, followed by S211N, S190N, and S389P, which was observed in 6 samples. All of these combinations were observed post-COVID-19 pandemic. This trend aligns with the findings of LaRinda et al., suggesting an increase in amino acid variations, specifically S211N, S190N, and S389P, in the RSV B F protein since 2020 [41]. Among the 28 post-COVID-19 pandemic RSV B samples, F protein variations were observed in 22 samples. This study confirms the frequent occurrence of variations in the F protein antigenic site, which is the target of recently approved RSV vaccines, in the aftermath of the COVID-19 pandemic. Therefore, continuous molecular genetic studies on RSV, including the RSV F protein, are deemed necessary. Additionally, further research is needed to understand the impact of amino acid variations in the RSV F protein on virus invasion and host immunity.

In South Korea, RSV occurrences typically begin in the fall, peak during the winter, and taper off in the spring, following a seasonal pattern. However, in the 2020/2021 season following the COVID-19 pandemic, no occurrences of RSV were reported. Subsequently, in the 2021/2022 season, after the relaxation of social distancing measures implemented due to the pandemic in October 2021, RSV cases started appearing, initially in December, and then explosively increasing until January of the following year. The incidence rate increased by approximately three-fold compared to pre-COVID-19 levels, a trend also observed in other studies [29,31,32,42].

The significant surge in RSV cases during South Korea’s 2021/2022 season is attributed to reduced exposure opportunities to RSV during the COVID-19 pandemic period due to social distancing measures, leading to a potential decline in immunity against RSV [43,44]. Additionally, in the 2022/2023 season, RSV occurrences exhibited a bimodal pattern and prolonged duration, a phenomenon also observed in studies conducted in countries like Australia and Japan [32,39]. Continuous monitoring of changes in RSV occurrence patterns post-COVID-19 pandemic is crucial, along with exploring response strategies. 

After the COVID-19 pandemic, changes in the age distribution of RSV recurrence were observed. Particularly, during the 2021/2022 season, not only did occurrences increase among children under 2 years old, but there was also a significant rise in RSV cases among children aged 2 to under 5 years old. Although RSV previously appeared to primarily affect infants, in the aftermath of the COVID-19 pandemic, RSV occurrences among children aged 2 to under 5 years old showed an increasing trend. During the period of approximately one year and 6 months with no RSV occurrences after the COVID-19 pandemic, infants under 2 years old had limited exposure to RSV, reducing their opportunities to develop immunity. In such circumstances, exposure to RSV in the post-COVID-19 pandemic period may have led to an increase in RSV occurrences among children aged 2 to under 5 years old. Similar findings have been observed in studies from Thailand, France, and Japan, among other countries [30,39,45]. The significant increase in perfusion of the RSV F IgG antibody levels in samples collected in 2021 compared to 2020, as reported by Reicherz et al., may help in understanding this phenomenon [44].

This study has several limitations. Firstly, the number of RSV samples before the COVID-19 pandemic was limited. Specifically, there was a shortage of suitable samples for the whole-genome analysis of RSV B from the 2019/2020 season, leading to limitations in the phylogenetic and mutational analysis of RSV B before and after COVID-19. Secondly, there was insufficient comparison data related to the clinical symptoms of RSV before and after COVID-19. Future research should investigate the relationship between the occurrence of RSV before and after COVID-19 and clinical symptoms.

Overall, the atypical resurgence of RSV after the COVID-19 pandemic is speculated to have been due to non-pharmacological interventions to control COVID-19, which reduced exposure to RSV and thereby lowered immunity to RSV. Additionally, this study observed an increase in the frequency of amino acid mutations in the F protein of RSV after the COVID-19 pandemic. The F protein is crucial as the target site for recently approved RSV vaccines. Therefore, molecular genetic analysis and continuous surveillance of RSV are necessary, as they can contribute to predicting and enhancing the effectiveness of vaccines.

## Figures and Tables

**Figure 1 viruses-16-00850-f001:**
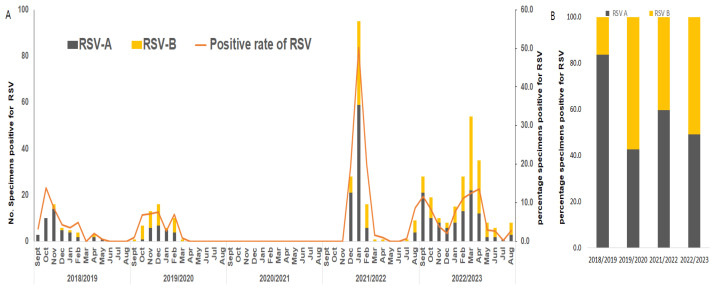
Distribution of RSV A and B in the Gwangju area, South Korea. (**A**) Monthly distribution of RSV A and B from the 2018/2019 to 2022/2023 seasons in the Gwangju area, South Korea. (**B**) The proportions of RSV A and B positive rates from the 2018/2019 to 2022/2023 seasons in the Gwangju area, South Korea.

**Figure 2 viruses-16-00850-f002:**
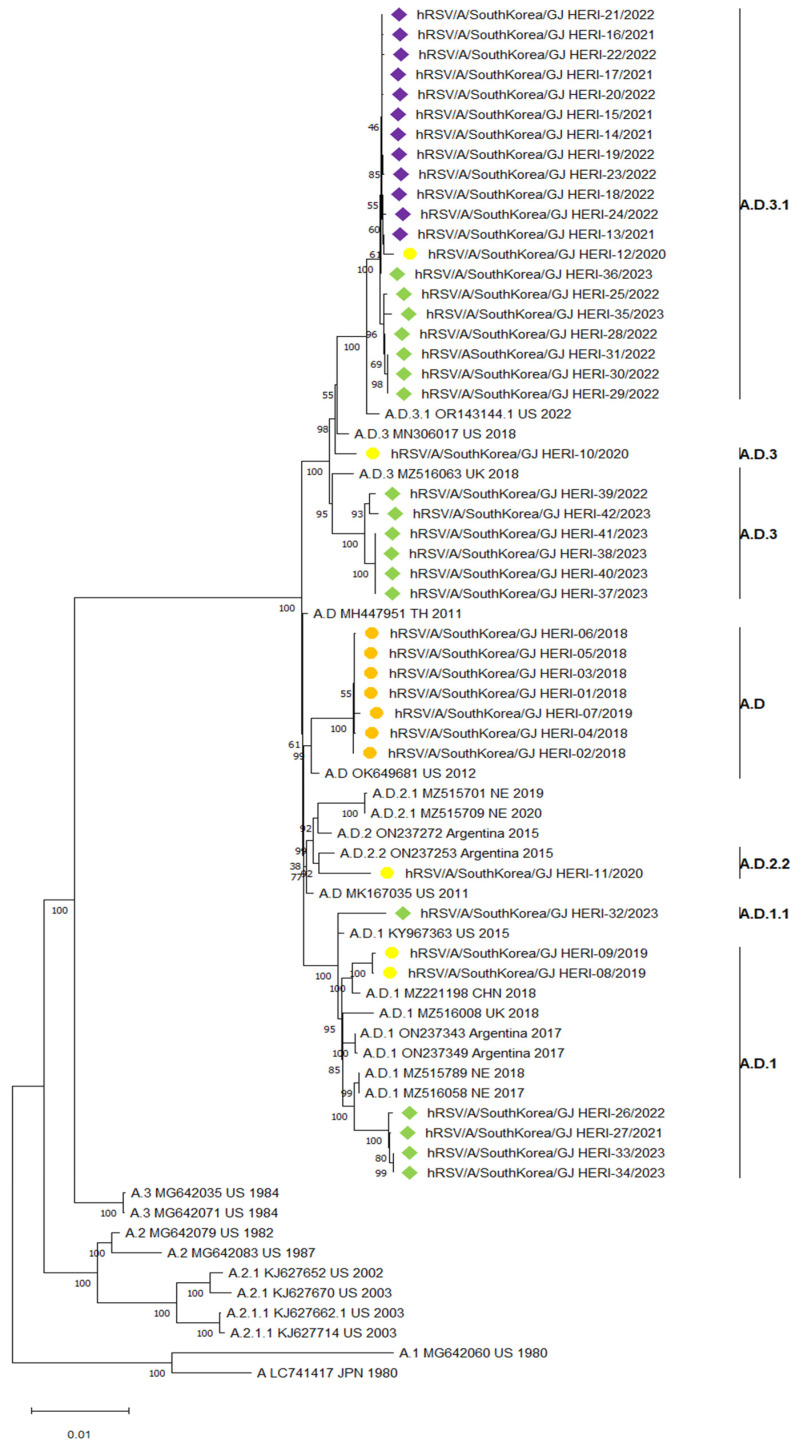
The phylogenetic tree was constructed based on 42 whole RSV A genome sequences. Samples before the COVID-19 pandemic are represented by circles, while samples after the COVID-19 pandemic are represented by diamonds. The 2018/2019 season is depicted by orange circles, the 2019/2020 season by yellow circles, the 2021/2022 season by purple diamonds, and the 2022/2023 season by green diamonds.

**Figure 3 viruses-16-00850-f003:**
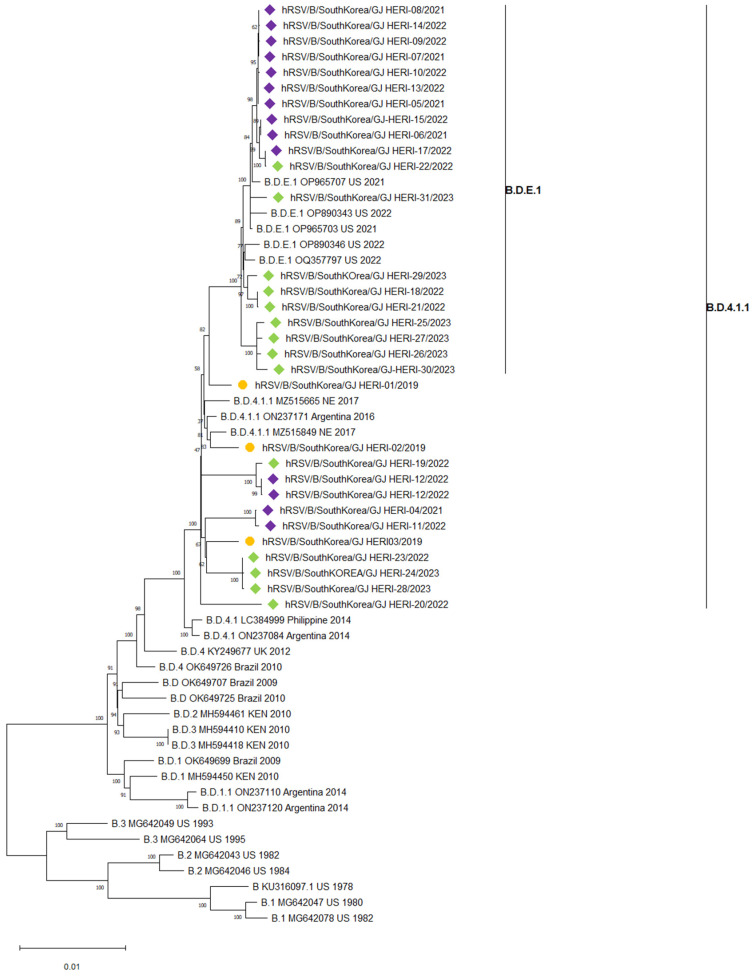
The phylogenetic tree was constructed based on 31 whole RSV B genome sequences. Samples before the COVID-19 pandemic are represented by circles, while samples after the COVID-19 pandemic are represented by diamonds. The 2018/2019 season is depicted by orange circles, the 2021/2022 season by purple diamonds, and the 2022/2023 season by green diamonds.

**Table 1 viruses-16-00850-t001:** Age and sex distribution of RSV positive samples from the 2018/2019 to 2022/2023 seasons in the Gwangju area, South Korea.

Season	2018/2019	2019/2020	2020/2021	2021/2022	2022/2023	*p*-Value ^1^
Total no. of samples analyzed	1248	1518	1194	1487	3470	
No. of RSV positive samples	49 (3.9%)	54 (3.6%)		151(10.2%)	220(6.3%)	<0.01 *
Gender						
Male	21 (42.9%)	28 (51.8%)	0	71 (47.0%)	124 (56.4%)	
Female	28 (57.1%)	26 (48.1%)	0	80 (53.0%)	96 (43.6%)	
Age						
0–2 years	15 (30.6%)	24 (44.4%)	0	65 (43.0%)	116 (52.7%)	<0.01 *
3–5 years	17 (34.7%)	17 (31.4%)	0	65 (43.0%)	66 (30.0%)	<0.01 *
6–10 years	6 (12.2%)	2 (3.7%)	0	7 (4.6%)	15 (6.8%)	
11–20 years	2 (4.1%)	1 (1.9%)	0	4 (2.6%)	7 (3.2%)	
21–40 years	2 (4.1%)	2 (3.7%)	0	4 (2.6%)	4 (1.8%)	
41–60 years	3 (6.1%)	3 (5.6%)	0	1 (0.7%)	4 (1.8%)	
60–90 years	4 (8.2%)	5 (9.3%)	0	5 (3.3%)	8 (3.6%)	

^1^ Chi-square test between 2019/2020 and 2021/2022 season. * *p* < 0.01.

**Table 2 viruses-16-00850-t002:** Amino acid substitutions in RSV A and B F protein antigenic sites found in genome sequences in the Gwangju area of South Korea, pre- and post-COVID-19 pandemic.

Antigen Site	Amino Acid Positions	RSV A Frequency No. (%)			RSV B Frequency No. (%)		
		Change	Pre COVID-19 (*n* = 12)	Post COVID-19 (*n* = 30)	Change	Pre COVID-19 (*n* = 3)	Post COVID-19 (*n* = 28)
Φ	62–96; 195–227				S211N		19 (67.8)
I	27–45; 312–318; 378–389	K42R		4 (13.3)	R42K		12 (42.8)
					S389P		18 (64.2)
					S389T		1 (3.5)
II	254–277	S276N	3 (25.0)	19 (63.3)	S276N		3 (10.7)
III	46–54; 301–311; 345–352; 367–378	-	-		-		-
IV	422–471				E463D		3 (10.7)
V	55–61; 146–194; 287–300				S190N		22 (78.5)

## Data Availability

All sequences from this study have been registered in the GISAID database (accession number: EPI_ISL19090938-19091010).

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
