# Peer review of "Impact of COVID-19 Pandemic Restrictions on Respiratory Virus Patterns: Insights from RSV Surveillance in Gwangju, South Korea"

_viruses, 2024, doi:10.3390/v16060850_

Round 1

Reviewer 1 Report

Comments and Suggestions for Authors

Dear authors,

I read with particular interest his manuscript, "Impact of COVID-19 Pandemic Restrictions on Respiratory Viruses Patterns: Insights from RSV Surveillance in Gwangju, South Korea." In it, the authors analyze the molecular epidemiology of RSV pre- and post-COVID-19 pandemic and how the social restrictions measures implemented due to the COVID-19 pandemic affected the pattern of RSV.

The paper is well-written and very complete. It is also a strong point of the article to discuss not only the G protein but also the F protein, which is of particular interest to us today. I would like to highlight some aspects or doubts that have arisen and could be clarified.

On page 2, line 79, it would be RSV instead of RSPV, wouldn't it? Otherwise, there would have to be a definition of the abbreviation.

Page 2, line 89: Were the samples taken from inpatients or outpatients? It is stated that the samples were taken from outpatients, but they were collected in hospitals; could this be clarified? What specific clinical symptoms should they have to be included in the study?

Page 3, line 101: Were the 73 samples sequenced all those with Ct≤25, or was it a selection?

Page 3, lines 138-139, The authors say that before the pandemic, the peak of RSV cases was between September and January, but looking at Figure 1A, cases start in September, but there is no peak in this month, the most pronounced peak is in November in the 2018/2019 season and in December in the 2019/2020 season. More detail could be provided.

No RSV cases were detected in the 2020/2021 season, but as shown in Table 1, many samples were tested during this season. The text indicates that no cases were detected, not because of a lack of testing but because there were no cases.

Page 4, line 143, December 2021, right?. The number of cases in February of the 2021/2022 season did not increase; it decreased significantly compared to January.

Since you are talking about incidence rates, please provide the incidence data. The following could also be included in Table 1

Page 4, line 169, 2021/2022 season? No cases in the 2020/2021 season.

Table 1 puts 0 cases in the No of RSV positive samples and by gender.

In figure captions 1 and 2, “The tree was created using the maximum likelihood method with a GTR+G+I substitution model and tested with 1,000 bootstrap replicates”, this is already indicated in the text: "All sequences from this study have been registered in the GISAID database (accession number: EPI_ISL1909096919091010)." Include this information in the text; do not include so much information in the figure caption.

The phylogenetic classification is based on the article by Goya et al., which is a preprint. Are there no other published reference articles?

The A.D.2.2 lineage was present before the pandemic but not detected afterwards. Did the authors see anything special about this lineage?

In the case of patient samples, the ethics committee that approved the study must be included in the text.

Thank you, and congratulations on your work

Author Response

Dear reviewer

Thank you for giving us the opportunity to submit a revised draft of the manuscript titled “Impact of COVID-19 Pandemic Restriction on Respiratory Virus Pattern: Insights from RSV Surveillance in Gwangju, South Korea” for publication in Viruses. We appreciate the time and effort that you and the reviewers dedicated to providing feedback on our manuscript and are grateful for the insightful comments on and valuable improvements to our paper. We have incorporated most of the suggestions made by the reviewers. Those changes are highlighted in blue. Please see below, in blue, for a point-by-point response to the reviewers’ comments and concerns.

Reviewer 2 Report

Comments and Suggestions for Authors

This study investigated surveillance data of RSV infections by conducting descriptive and molecular epidemiological analysis to understand the changes in the characteristics of RSV before and after the COVID-19 pandemic. I believe that the manuscript is generally well-written and provides detailed information about the study. Here, I have provided some minor comments aimed at improving the ease of understanding for readers.

Line 90: Please consider providing the population size of Gwangju so that the readers can understand the representativeness of this study in South Korea.

Section 2.1.: Please include the methods for descriptive analysis and statistical analysis for the contents of Table 1

Line 143, December 2022: December 2021?

Table 1: Please consider providing the proportion of positive samples for each gender and age (eg, for Male, 43% (21/49)).

Lines 191 and 194: I do not see the clade category of A.D.3 and A.D.1.1. in Figure 2 and 3, respectively. Should there be in the figures?

Figure 3: Should “B.D.4.1” be “B.D.4.1.1”?

Lines 243-244: if this sentence is before the explanation of Table 2 (lines 223-243), it is easier to follow the text contents.

Lines 229-230: Please consider providing the number of numerators and denominators and the proportion for easy understanding (ie, 25% (3/12), 63.3% (19/30), 4 cases among 30 samples (13.3%)).

Lines 234-235, R42K, S211N, S190N, and S389P: Are they a combination of mutations?

Liens 313-315: is this information in the result section?

Author Response

Manuscript Submission ID: 50c654fa4289c2b1af89c770c8d3d6f5

Response to Reviewer

Dear reviewer

Thank you for giving us the opportunity to submit a revised draft of the manuscript titled “Impact of COVID-19 Pandemic Restriction on Respiratory Virus Pattern: Insights from RSV Surveillance in Gwangju, South Korea” for publication in Viruses. We appreciate the time and effort that you and the reviewers dedicated to providing feedback on our manuscript and are grateful for the insightful comments on and valuable improvements to our paper. We have incorporated most of the suggestions made by the reviewers. Those changes are highlighted in blue. Please see below, in blue, for a point-by-point response to the reviewers’ comments and concerns.

Reviewers' Comments to the Authors:

Reviewer 2

This study investigated surveillance data of RSV infections by conducting descriptive and molecular epidemiological analysis to understand the changes in the characteristics of RSV before and after the COVID-19 pandemic. I believe that the manuscript is generally well-written and provides detailed information about the study. Here, I have provided some minor comments aimed at improving the ease of understanding for readers.

Line 90: Please consider providing the population size of Gwangju so that the readers can understand the representativeness of this study in South Korea.

We appreciate the valuable comments from the reviewer. The population size of Gwangju is approximately 1.5 million. Including this information will help readers better understand the representativeness of this study within South Korea. We have revised the content based on the reviewer's comments.

(page2, Line 87-89)

We, as a participating research institution in KINRESS, monitor acute respiratory infections (ARIs) in the Gwangju area, with a population of approximately 1.5 million.

Section 2.1.: Please include the methods for descriptive analysis and statistical analysis for the contents of Table 1

We really appreciate the reviewer’s comment on improving the quality of the manuscript. We revised the manuscript following the reviewer’s comments and suggestions. We have described the statistical analysis methods for descriptive data and group comparisons in lines 92-94.

(page3, Line 92-94)

Descriptive data were presented as frequency and percentage, and group comparisons were performed using the chi-square test. Results with a p-value less than 0.01 were considered significant.

Line 143, December 2022: December 2021?

Thank you for your careful review. “December 2021“is correct. I have revised line 147 to ”December 2021“

Table 1: Please consider providing the proportion of positive samples for each gender and age (eg, for Male, 43% (21/49)).

We thank the reviewer for this valuable comment, According to the suggestion from reviewer, we have added percentage values to Table 1.

Lines 191 and 194: I do not see the clade category of A.D.3 and A.D.1.1. in Figure 2 and 3, respectively. Should there be in the figures?

We thank the reviewer for this valuable comment. The 2019/2020 season had a total of 5 RSV A samples, which are indicated by yellow circles in Figure 2. Two samples belong to A.D.1, while the remaining three are distributed as follows: one in A.D.2.2, one in A.D.3, and one in A.D.3.1

Figure 3: Should “B.D.4.1” be “B.D.4.1.1”?

We appreciate the valuable comments from the reviewer. We have revised Figure 3 to reflect the change from B.D.4.1 to B.D.4.1.1.

Lines 243-244: if this sentence is before the explanation of Table 2 (lines 223-243), it is easier to follow the text contents.

We appreciate the valuable comments from the reviewer. In accordance with the reviewer’s comment, we have moved the content from lines 243-244 to lines 220-222. (page 8, Line 220-222)

Lines 229-230: Please consider providing the number of numerators and denominators and the proportion for easy understanding (ie, 25% (3/12), 63.3% (19/30), 4 cases among 30 samples (13.3%)).

We appreciate the valuable comments from the reviewer. In accordance with the reviewer’s comment, we have made revisions to the content in lines 227-231

(page8, Line227-231)

Although the sample sizes before and after COVID-19 were not equal, based on the analyzed samples, the frequency of the S276N mutation in the F protein of RSV A increased from 25% (3/12) to 63.3% (19/30) after the COVID-19 pandemic. The mutation K42R was observed in four cases among 30 samples (13.3%) after COVID-19.

Lines 234-235, R42K, S211N, S190N, and S389P: Are they a combination of mutations?

(page 8, Line 235)

We appreciate the valuable comments from the reviewer. In accordance with the reviewer’s comment, we have made revisions to the content in lines 235

Liens 313-315: is this information in the result section?

Thank you for reviewing to help me write a good paper. We noticed that the content was missing from the results section. We have incorporated your suggestions and included the relevant information in the results section.

(page8 Line 225-227)

Specifically, during the 2021/2022 season following the COVID-19 pandemic, the S276N amino acid mutation was observed in all 12 samples of the RSV A subgroup.